# Phage engineering to overcome bacterial Tmn immunity in *Dhillonvirus*

Wakana Yamashita[1,2], Kotaro Chihara [1], Aa Haeruman Azam[1], Kohei Kondo [3], Shinjiro Ojima[1], Azumi Tamura[1], Matthew Imanaka[2], Franklin L. Nobrega [4], Yoshimasa Takahashi [1], Koichi Watashi [1], Satoshi Tsuneda [2,5] & Kotaro Kiga [1,5,6] ✉

Bacteria possess numerous defense systems against phage infections, which limit phage infectivity and pose challenges for phage therapy. This study aimed to engineer phages capable of evading these defense systems, using the Tmn defense system as a model. We identified an anti-Tmn protein in the ΦSMS22 phage from the *Dhillonvirus* genus that inhibits Tmn function in *Escherichia coli*. Introducing this gene into the Tmn-sensitive ΦKSS9 phage enabled it to evade Tmn immunity. Additionally, we found that a single mutation in the *nmad5* gene, a DNA modification enzyme in *Dhillonvirus*, prevented Tmn from sensing phage infection. By mutating the *nmad5* gene in the Tmn-sensitive *Dhillonvirus*, we demonstrated that engineering phages to evade bacterial sensing mechanisms is another viable strategy. These two phage engineering approaches—introducing anti-defense genes and mutating sensing-related genes—present a promising strategy for establishing effective phage therapy by neutralizing bacterial defense systems.

The overuse of antibiotics has led to the emergence of drug-resistant bacteria, posing a global public health threat[1,2]. One promising alternative is phage therapy, which uses bacteriophages (phages) to target bacterial infections. Unlike antibiotics, phages have a unique bactericidal mechanism, enabling them to kill drug-resistant bacteria[3–5]. However, the widespread implementation of phage therapy has not yet been achieved. A significant obstacle is that bacteria inherently possess mechanisms to restrict phage infections[6,7]. For example, bacteria can alter or lose receptors necessary for phage entry, thereby preventing infection[8]. Additionally, bacteria restrict phage infection through defense systems such as CRISPR-Cas and restriction-modification systems[9]. Recent studies have revealed that bacteria employ a variety of defense systems to control phage infections, with over 78% of bacteria estimated to possess two or more defense systems[10].

To ensure that phage therapy is not hindered by bacterial receptor mutations, several approaches have been explored[11]. One approach involves combining phage therapy with antibiotics to prevent mutations in phage receptors[12]. While bacteria can mutate their membrane proteins to evade phage infection, by using phages targeting antibiotic efflux pumps, loss of such a protein comes at the cost of antibiotic susceptibility[13]. Another method involves using a pool of phages with randomly modified tail genes[14]. This ensures that even if mutations occur in the phage receptor, there will be

phages in the pool capable of infecting the bacteria, thus maintaining bactericidal activity. Although the engineering of phage tails targeting bacterial receptors has progressed, the development of phages designed to evade bacterial defense systems has been limited[15–19].

In this study, we focus on the Tmn defense system, a member of the KAP family NTPase characterized by two unusual transmembrane domains. While the system can inhibit propagation of T2, P1, phiVi, and phiX, the precise defense mechanism of Tmn remains unexplored[20,21]. In this study, we selected the Tmn defense system as a model to design synthetic bacteriophages capable of overcoming bacterial defense systems with unknown mechanisms.

## Results

### Identification of anti-Tmn

In our previous work, we demonstrated that *Dhillonvirus* phages ΦSMS22 and ΦSMA8 exhibit distinct infectivity profiles against the Tmn defense system[22] (Fig. 1a). ΦSMS22 evaded the Tmn defense mechanism, whereas ΦSMA8 was inhibited by Tmn, resulting in lysis from without (Fig. 1b and Supplementary Fig. 1a and Supplementary Data 1). These findings suggest that ΦSMS22 either lacks genes triggering the Tmn response or possesses genes enabling it to evade Tmn. To investigate the possibility of *anti-Tmn*

[1]Research Center for Drug and Vaccine Development, National Institute of Infectious Diseases, Tokyo, 162-8640, Japan. [2]Department of Life Science and Medical Bioscience, Waseda University, 2-2 Wakamatsu-cho, Shinjuku-ku, Tokyo, 162-8480, Japan. [3]Antimicrobial Resistance Research Center, National Institute of Infectious Diseases, Tokyo, Japan. [4]School of Biological Sciences, University of Southampton, Southampton, SO17 1BJ, UK. [5]Phage Therapy Institute, Comprehensive Research Organization, Waseda University, 2-2 Wakamatsu-cho, Shinjuku-ku, Tokyo, 162-8480, Japan. [6]Division of Bacteriology, Department of Infection and Immunity, School of Medicine, Jichi Medical University, Shimotsuke-shi, Tochigi, 329-0498, Japan. ✉e-mail: k-kiga@niid.go.jp

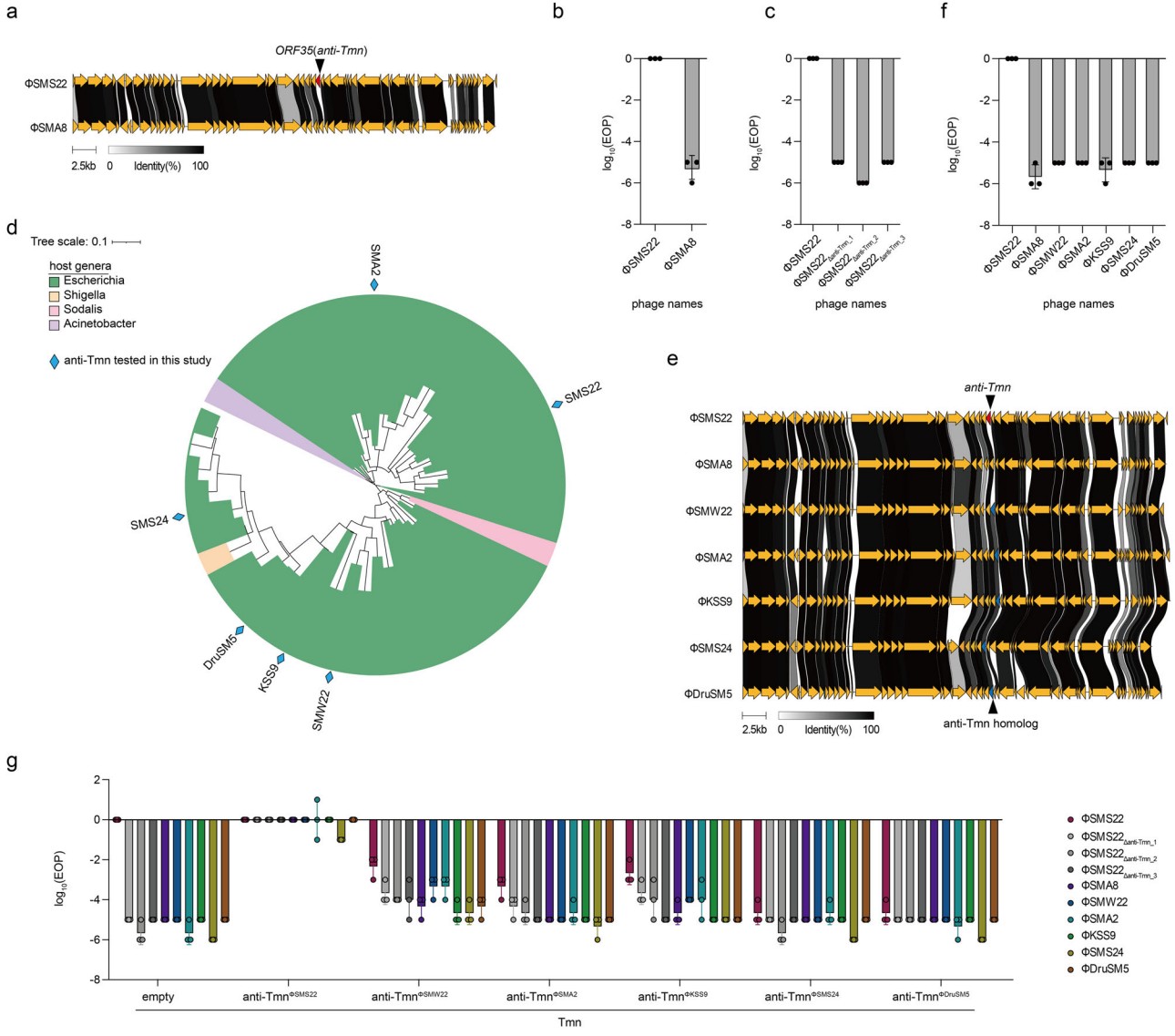

**Fig. 1 | Identification of phage genes involved in Tmn evasion. a** Genomic comparison of ΦSMS22 and ΦSMA8. **b** Bar graph summarizing the results of spot assays assessing the infectivity of phages ΦSMS22 and ΦSMA8 on bacterial strains equipped with the Tmn defense system. The y-axis represents the logarithm of the efficiency of plating (EOP), calculated as the ratio of PFU/mL for bacteria carrying the Tmn defense system to the PFU/mL for bacteria carrying empty vectors. The x-axis indicates the names of the phages. The experiment was performed in triplicate, and the bar graph shows the mean values with error bars representing standard deviations. **c** Bar graph based on the measurement of phage infectivity against Tmn defense system by deletion of *ORF35* (*anti-Tmn*) in ΦSMS22 phage. The deletion mutant phages were synthesized using the in vitro synthesis method, and three independent phages (ΦSMS22$_{\Delta anti-Tmn\_1}$, ΦSMS22$_{\Delta anti-Tmn\_2}$, ΦSMS22$_{\Delta anti-Tmn\_3}$) were obtained after reboot and used for spot assay. **d** Phylogenetic tree of anti-Tmn homologs from diverse bacterial genera (GA: 20, coverage > 40%, identity ≤95%). **e** Genomic comparison of ΦSMS22, ΦSMA8, ΦSMW22, ΦSMA2, ΦKSS9, ΦSMS24, and ΦDruSM5. **f** Bar graph summarizing the results of a phage spot assay assessing the infectivity of the seven phages against bacteria with Tmn defense system. **g** Bar graph based on spot assay of phage on bacteria with Tmn and anti-Tmn or anti-Tmn homologs.

genes in ΦSMS22, we first knocked out *ORF35*, a gene present only in ΦSMS22 and absent in ΦSMA8. Deletion of *ORF35* from the ΦSMS22 genome resulted in reduced infectivity against *E. coli* expressing Tmn, suggesting that *ORF35* functions as an anti-Tmn (Fig. 1c and Supplementary Fig. 1b and Supplementary Data 1).

We utilized AlphaFold to predict the structure of the anti-Tmn protein. The predicted structure revealed that anti-Tmn adopts a rod-shaped configuration, characterized by long and short N-terminal α-helixes aligned with each other, with a C-terminal three-stranded β-sheet sandwiched between two short α-helixes[23] (Supplementary Fig. 2). The N- and C-terminal secondary structures are connected by a 12-amino acid disordered linker. Notably, AlphaFold predicted a higher pLDDT value for the

anti-Tmn dimer compared to the monomer (pTM = 0.55 for dimer vs. pTM = 0.45 for monomer) (Supplementary Fig. 2). Moreover, the AlphaFold prediction for the anti-Tmn dimer yielded high-confidence predictions for both N- and C-terminal structures between two protomers (Supplementary Fig. 2).

We analyzed the conservation of anti-Tmn across all bacteriophage and archaeal virus proteins available in the GenBank database. To do this, we built a Hidden Markov Model (HMM) profile for anti-Tmn based on clustered representatives from the BLASTp results (see Method). Applying a low threshold (GA: 20 and coverage >40%), we identified 85 hits. From these, we generated a phylogenetic tree using 48 non-redundant amino acid sequences (identity≤95%), with the majority (45 out of 48) originating from

coliphages (Fig. 1d). Interestingly, anti-Tmn homologs were exclusively found within *Dhillonvirus*, except for a single occurrence in *Skarprettervirus*, indicating that anti-Tmn is a specialized system confined to specific viral lineages.

Subsequently, we searched for phages possessing anti-Tmn homologs in our phage library and identified five phages: ΦSMW22, ΦSMA2, ΦKSS9, ΦSMS24, and ΦDruSM5 (Fig. 1e). Like ΦSMS22 and ΦSMA8, these phages all belong to the *Dhillonvirus* genus. Interestingly, despite possessing anti-Tmn homologs, these five phages were still inhibited by Tmn (Fig. 1f and Supplementary Fig. 3 and Supplementary Data 1). To investigate this phenomenon, we cloned the *anti-Tmn* homologs from all six phages into plasmids under an arabinose-inducible promoter. We co-expressed the anti-Tmn homologs with Tmn in *E. coli* and challenged each strain against seven phages (Fig. 1g and Supplementary Fig. 4 and Supplementary Data 1). All phages inhibited by Tmn regained infectivity when co-expressed with the ΦSMS22-derived anti-Tmn. The ΦSMS22-derived anti-Tmn exhibited strong inhibitory activity against Tmn from PLG026[21] (Addgene plasmid # 157904), whereas the anti-Tmn homologs from ΦSMW22 and ΦKSS9 showed limited effects (Fig. 1g and Supplementary Fig. 4). Additionally, other anti-Tmn homologs did not exhibit inhibitory activity against Tmn derived from PLG026, despite their high amino acid sequence similarity (Fig. 1g and Supplementary Fig. 4 and Supplementary Data 1). These results suggest that even slight variations in amino acid sequences can influence the efficacy of anti-Tmn.

Next, we examined whether anti-Tmn could counteract Tmn systems other than PLG026. From the 397 Tmn proteins registered in the Bacterial RefSeq database[24,25], 81 non-redundant Tmn homologs, clustered at 95% sequence identity, were identified. Among them, 51 Tmn homologs were

from Gammaproteobacteria (Supplementary Fig. 7). Four Tmn variants were cloned from clinical isolates—two from *Escherichia coli*, one from *Enterobacter cloacae*, and one from *Klebsiella pneumoniae*—and co-expressed with anti-Tmn in *E. coli* DH10B. We then infected the strains with *Dhillonvirus* phages to assess whether anti-Tmn could inhibit various Tmn systems. TmnA-JBABADF-19-0057, a Tmn variant that strongly blocked ΦSMW22 infection, was effectively counteracted by all anti-Tmn homologs (Supplementary Fig. 8 and Supplementary Data 1). Another variant, TmnA-JBABADI-19-0057, inhibited a broad range of *Dhillonvirus* phages, and its defensive activity was neutralized by anti-Tmn[ΦSMS22] and anti-Tmn[ΦDruSM5]. These findings suggest that each anti-Tmn targets a specific Tmn. Furthermore, PLG026-derived Tmn was found to strongly block ΦT1 infection, a *Tunavirus* (Supplementary Fig. 9 and Supplementary Data 1). Anti-Tmn[ΦSMS22], anti-Tmn[ΦSMW22], anti-Tmn[ΦSMA2], and anti-Tmn[ΦKSS9] neutralized this defense activity, with anti-Tmn[ΦSMS22] showing the strongest effect. Overall, these results demonstrate that anti-Tmn has the potential to inhibit various Tmn systems and restore the infectivity of multiple phages.

## Synthesis of phages evading Tmn defense system using anti-Tmn

Having identified a potent anti-Tmn protein in ΦSMS22, we aimed to engineer phages that could evade the Tmn defense system. We selected ΦKSS9 for further study as it failed to counteract the defensive activity of Tmn despite encoding an *anti-Tmn* homolog. We therefore replaced the *anti-Tmn* gene in ΦKSS9 with the one from ΦSMS22 using in vitro assembly[26,27] (Fig. 2a). Three independent plaques (ΦKSS9[anti-Tmn_1], ΦKSS9[anti-Tmn_2], and ΦKSS9[anti-Tmn_3]) were selected and propagated. The

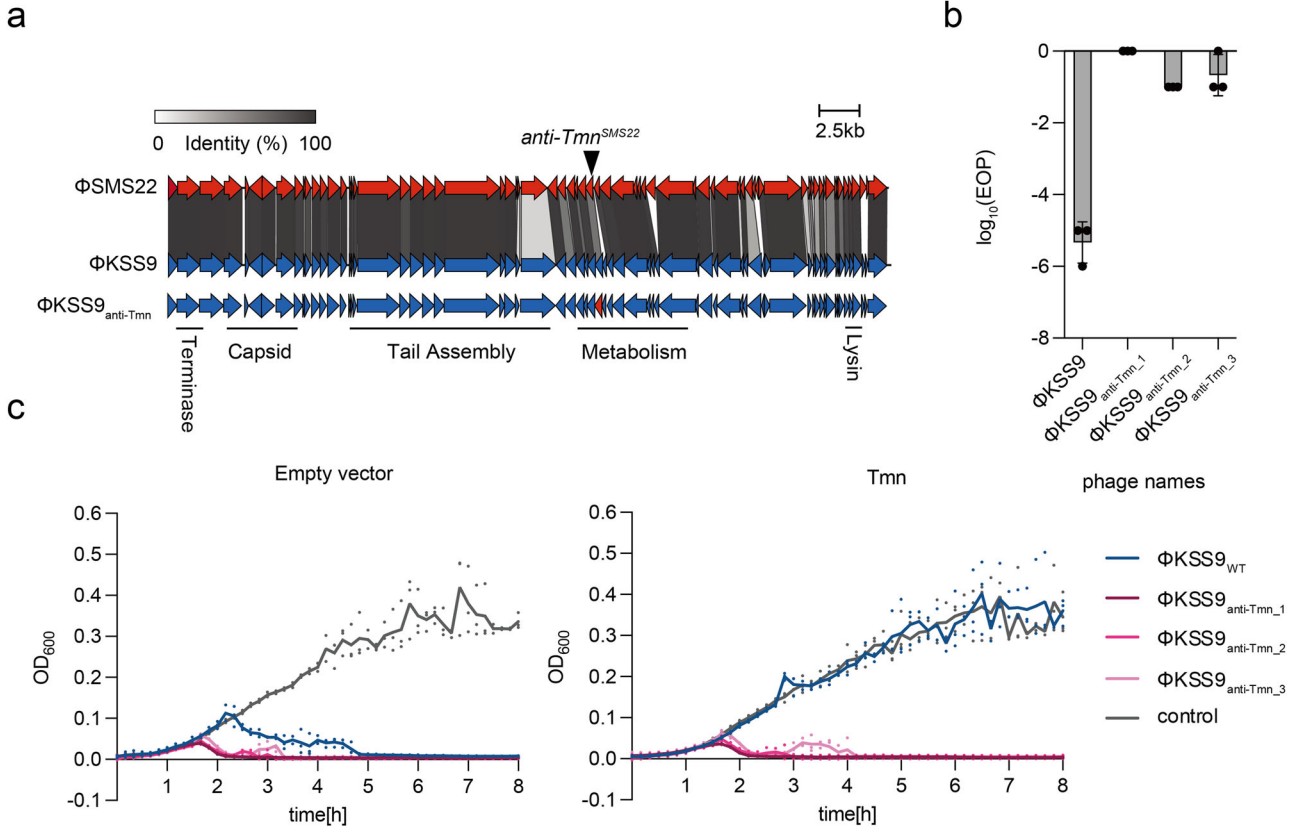

**Fig. 2 | Modification of anti-Tmn on ΦKSS9. a** Genome map of ΦKSS9[anti-Tmn]. The *anti-Tmn* homolog carried by ΦKSS9 was replaced with the *anti-Tmn* carried by ΦSMS22 (ΦKSS9[anti-Tmn]). **b** Bar graph based on measurement of phage infectivity against Tmn defense system by insertion of *anti-Tmn* in ΦKSS9 phage (ΦKSS9[anti-Tmn]). Phage infectivity was measured using the spot assay. The experiment was performed in triplicate, and the bar graph shows the mean values with error bars representing standard deviations. **c** Comparison of bactericidal activity of ΦKSS9[WT] and ΦKSS9[anti-Tmn]. Phages were inoculated to DH10B expressing Tmn at MOI 5 × 10^{-4}, and bacterial growth was measured. Each experiment was done in triplicate, with the average represented as a line graph and raw data points plotted.

wild-type ΦKSS9 strain was susceptible to Tmn immunity, whereas the ΦKSS9_anti-Tmn showed resistance to Tmn immunity (Fig. 2b and Supplementary Fig. 10 and Supplementary Data 1). To investigate if ΦKSS9_anti-Tmn could overcome Tmn immunity in liquid cultures, we inoculated Tmn-expressing bacteria with the engineered phages at an MOI of $5 \times 10^{-4}$ and monitored bacterial growth over 8 h (Fig. 2c and Supplementary Data 1). Tmn-expressing strains inoculated with ΦKSS9 showed growth curves similar to those of Tmn-expressing strains without inoculation, indicating that ΦKSS9 could not overcome Tmn immunity in liquid culture. In contrast, the growth of Tmn-expressing strains inoculated with ΦKSS9_anti-Tmn was rapidly suppressed (Fig. 2c and Supplementary Data 1). These findings highlight the potential of engineering phages with immune evasion proteins to bypass bacterial defense systems.

## Isolation of phages escaping Tmn immunity

Identifying defense system inhibitors requires screening through generations of phage mutants or constructing phage gene expression libraries[22]. Therefore, we attempted to construct phages capable of evading the defense system through genetic mutations. Host defense systems are activated upon detection of phage infection or phage-derived genes, but phages can evade this detection by acquiring spontaneous mutations in their genome[16,28]. As escape phages against Tmn have not been reported, we initially aimed to obtain phages capable of escaping from Tmn by co-cultivating Tmn-expressing bacteria with phages sensitive to Tmn. We used ΦSMA8 (originally lacking the *anti-Tmn* homolog) and ΦSMS22_Δanti-Tmn, which are sensitive to Tmn immunity (Fig. 3a). Escape phages capable of infecting Tmn-expressing strains were

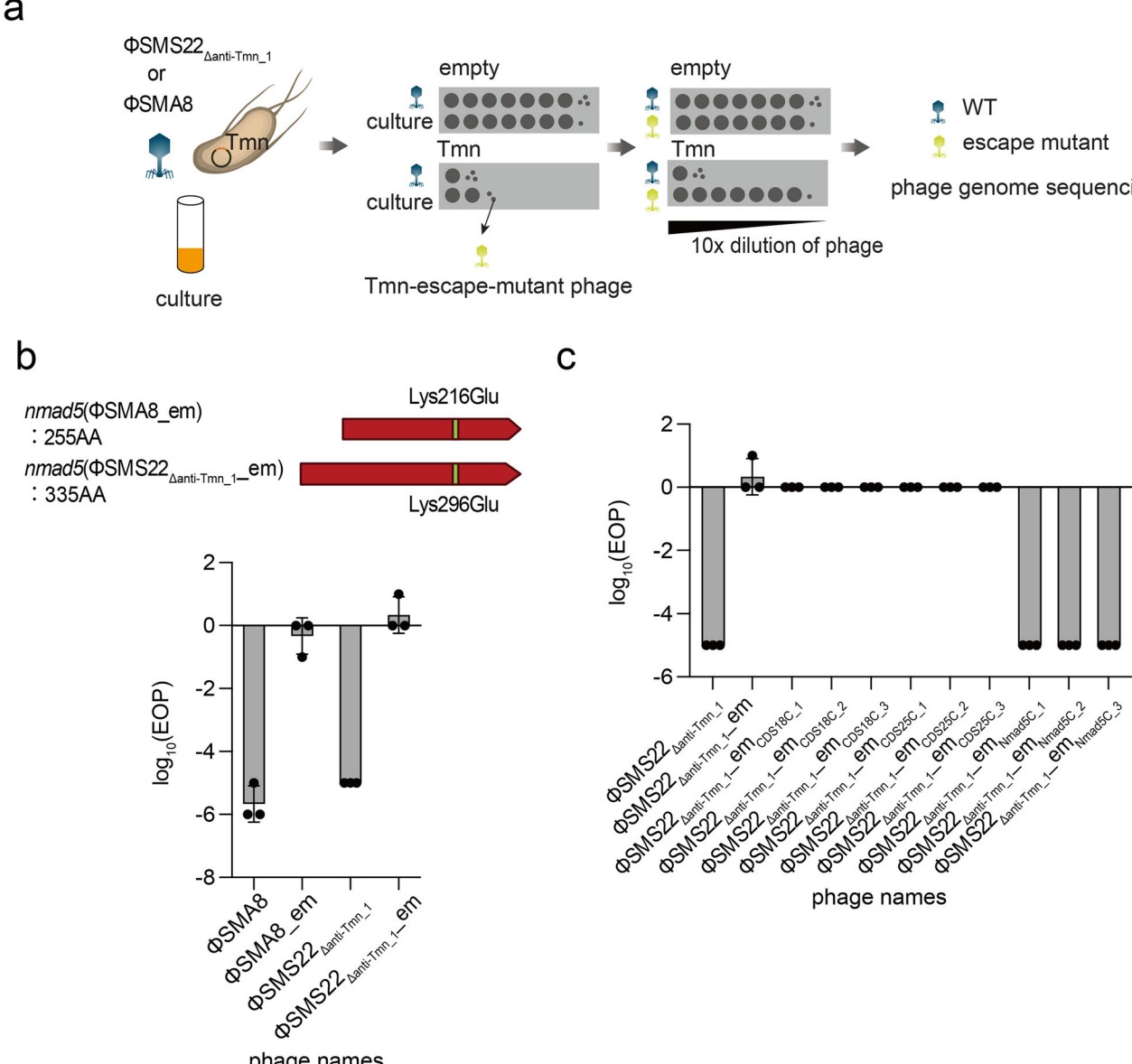

Fig. 3 | Identification of phage genes involved in Tmn activation. a Isolation of phages that escaped Tmn immunity. ΦSMA8 or ΦSMS22_Δanti-Tmn was co-cultured overnight in a liquid medium with bacteria harboring Tmn. Escape mutants were isolated from the respective co-culturing. The genomes of the escape mutants were analyzed and mapped to the wild-type genome. The genes with common mutations identified in both escape phages were considered potential factors that activate the Tmn defense system. b Bar graph based on spot assay results for infectivity of escape mutant phage against Tmn-carrying strains. Escape mutants avoided infection inhibition by Tmn. Two other phage-derived escape mutant phages carried the same mutation at the same position in *nmad5*. The experiment was performed in triplicate, and the bar graph shows the mean values with error bars representing standard deviations. c Bar graph based on spot assay to identify the Tmn trigger. Three genes that contained mutations in ΦSMS22_Δanti-Tmn escape mutant phage were reverted to the WT sequence one by one. When *nmad5* was returned to WT, ΦSMS22_Δanti-Tmn became inhibited from infection by Tmn.

**Table 1 | Genetic mutations carried by escape mutant phage**

| Phage name | Mutated gene | Gene role | Mutation |
|---|---|---|---|
| ΦSMA8_em | CDS7 | Nmad5 | Lys216Glu |
| | CDS9 | Head scaffolding protein | Lys126Thr |
| | CDS67 | Major tail protein | Pro53fs |
| ΦSMS22$_{\Delta anti\text{-}Tmn}$_em | CDS6 | Nmad5 | Lys296Glu |
| | CDS18 | Hypothetical protein | Pro59fs, Pro60-Ter66del |
| | CDS25 | Tail protein | Asp1098fs, Gln1104fs |

**Table 2 | Genetic mutations in synthetic phages used in the search for the Tmn trigger**

| Phage name | Sensitivity | Mutated gene | | |
|---|---|---|---|---|
| | | CDS6 (nmad5) | CDS18 | CDS25 |
| ΦSMS22$_{\Delta anti\text{-}Tmn}$ | Sensitive | | | |
| ΦSMS22$_{\Delta anti\text{-}Tmn}$_em | Resistant | Lys296Glu | Pro59fs, Pro60-Ter66del | Asp1098fs, Gln1104fs |
| ΦSMS22$_{\Delta anti\text{-}Tmn}$_em$_{Nmad5C}$ | Sensitive | | Pro59fs, Pro60-Ter66del | Asp1098fs, Gln1104fs |
| ΦSMS22$_{\Delta anti\text{-}Tmn}$_em$_{CDS18C}$ | Resistant | Lys296Glu | | Asp1098fs, Gln1104fs |
| ΦSMS22$_{\Delta anti\text{-}Tmn}$_em$_{CDS25C}$ | Resistant | Lys296Glu | Pro59fs, Pro60-Ter66del | |

obtained for both phages and named SMA8_em and SMS22$_{\Delta ant\text{-}Tmn}$_em, respectively (Fig. 3b and Supplementary Fig. 11a and Supplementary Data 1). Whole-genome sequence analysis revealed that both escape mutant phages had three mutations each, with a common mutation occurring in the gene encoding the DNA modification enzyme *nmad5*, resulting in an amino acid substitution of K216E of ΦSMA8 and K296E in ΦSMS22, respectively (Fig. 3b and Table 1). Despite the mutations occurring at different sites, K216E in ΦSMA8 and the K296E in ΦSMS22, their relative positions within *nmad5* are equivalent. This is because *nmad5* in ΦSMA8 has a truncated N-terminus compared to *nmad5* in ΦSMS22, suggesting similar functions (Fig. 3b). To confirm which of the mutated genes is responsible for Tmn immunity, we reverse mutated each gene to the original sequence one by one in ΦSMS22$_{\Delta anti\text{-}Tmn}$ using the in vitro rebooting approach. The synthesized phages were subject to plaque assay against either control or Tmn-expressing bacteria (Table 2). A mutation in *CDS18* or *CDS25*, identified in the escape phage aside from *nmad5* (Table 1), did not affect susceptibility to Tmn when reverted (Fig. 3c and Supplementary Fig. 11b and Supplementary Data 1). However, restoring the *nmad5* mutation in ΦSMS22$_{\Delta anti\text{-}Tmn}$_em recovered susceptibility to Tmn immunity. These results indicate a potential role of *nmad5* in Tmn immunity. Attempts to clone *nmad5* were unsuccessful due to its apparent toxicity (Supplementary Fig. 12). Notably, the deletion of *nmad5* from ΦSMS22$_{\Delta anti\text{-}Tmn}$ did not abolish the susceptibility to Tmn-mediated infection inhibition (Supplementary Fig. 13). These results suggest that phages obtain the ability to escape from Tmn defense systems by acquiring a single missense mutation in *nmad5*. As *nmad5* is predicted to be involved in the biosynthesis of modified DNA bases in prokaryotes[29], Tmn may be able to interfere with infection by phages whose DNA modifications are facilitated by *nmad5*.

## Synthesis of phages evading Tmn defense system utilizing a *nmad5* mutation

As Tmn likely detects DNA modifications from *nmad5*, we attempted to construct a phage capable of evading Tmn detection by mutating the gene. We constructed the genome of ΦSMS22$_{\Delta anti\text{-}Tmn\_1}$_Nmad5$_{K296E}$ by in vitro assembly such that the K296E mutation was introduced into *nmad5* of ΦSMS22$_{\Delta anti\text{-}Tmn\_1}$ (Fig. 4a). Three synthetic phages were obtained from independent plaques of ΦSMS22$_{\Delta anti\text{-}Tmn}$_Nmad5$_{K296E}$ and were used to infect Tmn-expressing *E. coli*. We found that ΦSMS22$_{\Delta anti\text{-}Tmn}$ was susceptible to Tmn immunity, but ΦSMS22$_{\Delta anti\text{-}Tmn\_1}$_Nmad5$_{K296E}$ showed strong resistance to Tmn defense (Fig. 4b and Supplementary Fig. 14 and Supplementary Data 1). To evaluate whether the synthesized ΦSMS22$_{\Delta anti\text{-}Tmn}$_Nmad5$_{K296E}$ could overcome Tmn immunity in liquid culture, Tmn-expressing bacteria was inoculated with ΦSMS22$_{\Delta anti\text{-}Tmn}$_Nmad5$_{K296E}$, and bacterial growth was monitored. ΦSMS22$_{\Delta anti\text{-}Tmn}$ was susceptible to Tmn immunity, whereas ΦSMS22$_{\Delta anti\text{-}Tmn}$_Nmad5$_{K296E}$ rapidly inhibited the growth of Tmn-expressing strains (Fig. 4c and Supplementary Data 1). These results suggest that introducing mutations in *nmad5* of *Dhillonvirus* is a viable strategy to overcome the Tmn defense system.

## Discussion

Understanding the strategies employed by phages to counteract host defense systems is crucial for improving phage therapy outcomes. This study successfully engineered synthetic phages to overcome the Tmn defense system using two approaches: countering the Tmn defense system by introducing an *anti-Tmn* gene and avoiding the Tmn-defense system by mutating the gene implicated in phage detection (Fig. 4d)[30,31].

We identified several anti-Tmn proteins, with the one from ΦSMS22 exhibiting the strongest inhibitory activity against Tmn from PLG026. Interestingly, five phages similar to ΦSMS22, despite possessing *anti-Tmn* homologs, remained susceptible to Tmn. The lack of inhibition despite homology suggests countering Tmn defense is highly dependent on the specific amino acid sequence of anti-Tmn proteins. Additionally, four Tmn variants cloned in this study exhibited distinct phage defense activities. The defense activity of TmnA-JBABADF-19-0057 was inhibited by all tested anti-Tmn proteins and the defense activity of TmnA-JBABADI-19-0057 was predominantly suppressed by anti-Tmn$^{ΦSMS22}$ and anti-Tmn$^{ΦDruSM5}$. These findings suggest that anti-Tmn proteins exhibit specificity in their ability to counteract different Tmn systems. We also demonstrated that the efficacy of anti-Tmn in overcoming Tmn-mediated defense against the T1 phage, which belongs to a phage group other than *Dhillonvirus*. This finding suggests that anti-Tmn could potentially rescue not only *Dhillonvirus* phages but also a broader range of phages from the Tmn defense system.

Another approach to evade the Tmn defense system involved inducing a single gene mutation in *Nmad5*, a gene responsible for nucleotide modification[29]. Numerous studies have reported associations between phage defense systems and various DNA modifications[32,33]. Similarly, in the case of the Tmn defense system, DNA modifications may also influence its defensive activity. Intriguingly, deletion of *nmad5* did not result in the evasion of Tmn inhibition. This suggests that the single mutation in *nmad5* observed in the Tmn-escape phages does not merely disrupt the function of the wild-type *nmad5*. Instead, it is possible that this single mutation imparts a novel function to *nmad5*, which facilitates evasion of the Tmn defense system. These findings may provide valuable insights into the mechanisms of the Tmn defense system; however, further studies are needed to elucidate the detailed mechanisms.

We engineered synthetic phages to evade the Tmn defense system using two strategies: introducing an *anti-Tmn* gene and inducing a mutation in *nmad5*, a gene involved in sensing phage infection by Tmn. Each approach offers unique advantages and drawbacks. Identifying these genes can be time-consuming, often requiring techniques like constructing phage gene expression libraries[17], creating a phage knockout library[22], or using chimeric phages[34]. However, once identified, incorporating the anti-defense gene into the phage genome becomes a

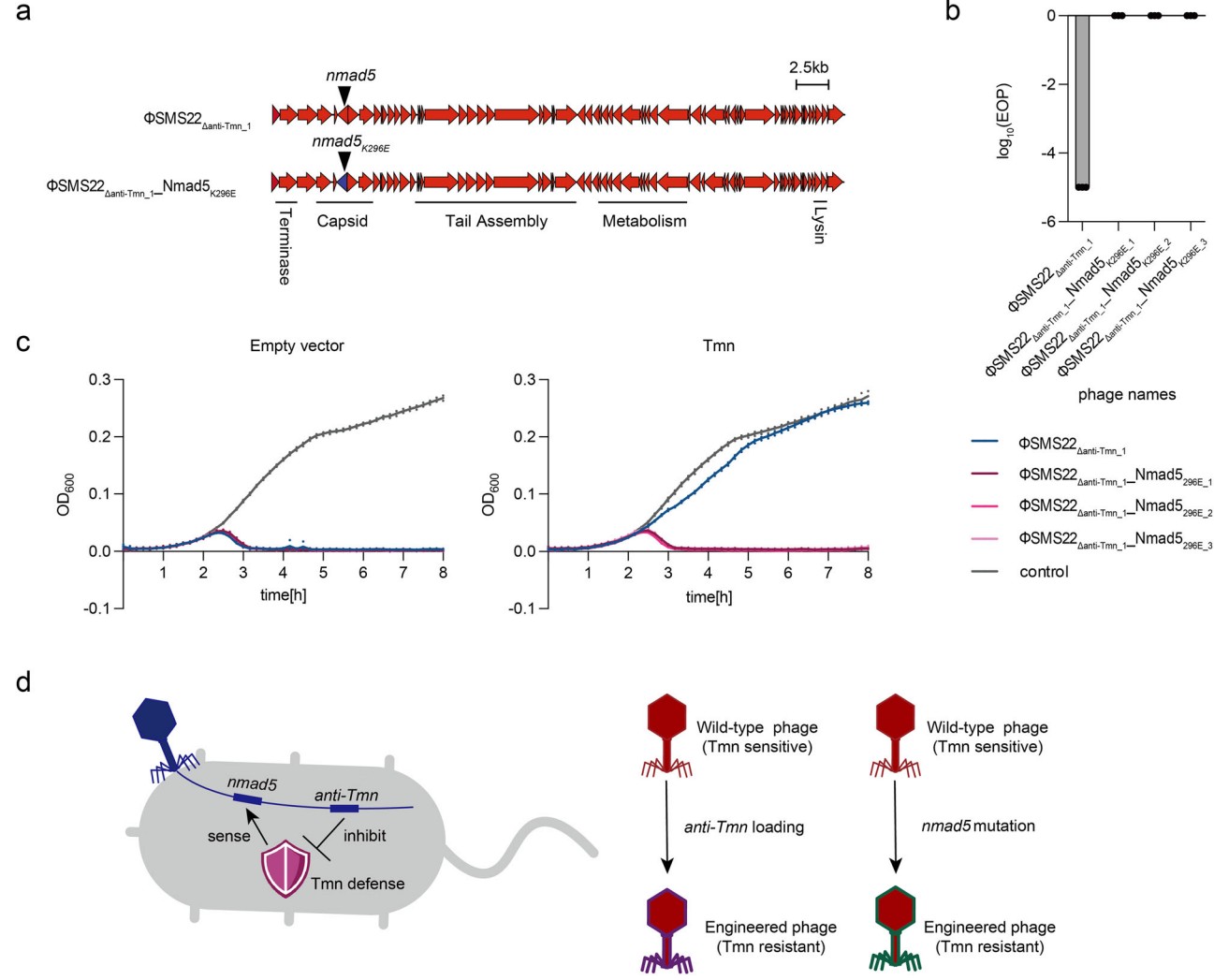

**Fig. 4 | Mutation of Tmn activator *nmad5* in ΦSMS22$_{\Delta anti-Tmn}$. a** Genomic map of ΦSMS22$_{\Delta anti-Tmn}$. *nmad5* is located near the capsid gene on the ΦSMS22$_{\Delta anti-Tmn}$ genome. **b** Bar graph based on measurement of phage infectivity against Tmn defense system by single amino acid substitution of *nmad5* in ΦSMS22$_{\Delta anti-Tmn}$. Phage infectivity was measured using a spot assay. The experiment was performed in triplicate, and the bar graph shows the mean values with error bars representing standard deviations. **c** Comparison of bactericidal activity of ΦSMS22$_{\Delta anti-Tmn}$ and ΦSMS22$_{\Delta anti-Tmn}$_Nmad5$_{K296E}$. Each phage was inoculated with Tmn-expressing DH10B at a MOI 5× 10$^{-4}$ and bacterial growth was measured. The experiment was conducted with *n* = 3. **d** Overview of this study. Phages were engineered with inhibitors or by introducing mutations in *nmad5* to escape Tmn-mediated infection inhibition.

straightforward method for overcoming specific defense systems. Isolating phages with mutations in the gene sensed by the defense system generally requires less time and effort compared to finding an anti-defense system. However, this approach has limitations. Mutations in essential genes, such as those encoding capsid components, are sometimes difficult to introduce and maintain[16]. Even when such mutations confer immune evasion, they may come at a fitness cost, reducing phage replication efficiency and limiting their utility in phage therapy. Furthermore, this strategy is limited because mutating a specific sensor gene is not effective for phages that do not carry that sensor gene.

In phage therapy, matching phage to target bacteria is critical. However, this process is often time-consuming and labor-intensive, as bacteria possess a variety of defense systems to counteract phage invasion[35]. Therefore, more studies on host defense and phage anti-defense systems are needed to systematically build phages that counter the defense systems. In this study, we used Tmn as a model system and demonstrated two strategies to overcome this defense by using synthetic phages. These are expected to become a rational and rapid method for matching phage and target bacteria in phage therapy in the future.

## Methods

### Culture Conditions for Bacterial Strains

The bacterial strains used in this study are listed in Supplementary Table 1. All bacteria in this study were *E. coli* DH10B. A single bacterial colony was inoculated into 2 mL of Luria-Bertani (LB) medium (BD Difco, Franklin Lakes, NJ, USA) and incubated overnight at 37 °C with shaking at 200 rpm. Chloramphenicol (Cm) was added to the LB when culturing bacteria expressing Tmn, and Cm and carbenicillin (Crb) were added to the LB when culturing bacteria co-expressing Tmn and anti-Tmn. Antibiotics were added to the growth medium at the final concentration of 100 μg/mL for carbenicillin (Crb) and 34 μg/mL for chloramphenicol (Cm).

### Spot assay (2-layer agar method)

For the top agar, 300 μL of the overnight culture was added to 10 mL of LB top agar (LTA) containing 0.5% agarose and 1 mM CaCl$_2$. For induction of the anti-Tmn, 100 μL of 0.2 wt% arabinose was added to LTA mixes. Molten solutions were poured on LB agar plates and allowed to set. Plates were spotted with 2 μL of 10-fold serial phage dilutions, incubated overnight at 37 °C, and then the plaques were counted.

## Phage Synthesis

Phages were synthesized using an in vitro synthesis method. Phage genomic DNA fragments and the insertion sites of interest were amplified from the phage genome using PCR. PCR was performed using KOD FXNeo (TOYOBO) or KOD One® PCR Master Mix (TOYOBO). PCR enzyme and primers listed in the Supplementary Table 2 with the PCR conditions described in Supplementary Table 3. DNA amplified fragments were assembled using NEBuilder HiFi DNA Assembly (New England Biolabs, Inc., Ipswich, MA, USA) by incubation at 50 °C for 1.5 h. Assembled genomes were electroporated into *E. coli* HST08 Premium Electro-Cells (Takara Bio Inc., Shiga, Japan) using ELEPO21 (Nepa Gene Co., Ltd., Chiba, Japan). After electroporation, 1 mL of SOC medium (Takara Bio Inc.) was added to the samples and incubated at 37 °C for 30 min with shaking at 200 rpm. The samples were then added to 3 mL of LTA, spread on LB plates, and incubated at 37 °C overnight. The next day, three plaques were collected from each plate and each was added to DH10B overnight culture medium diluted 100-fold in LB. These samples were incubated at 37 °C, 200 rpm until the solution became clear, and the solution was filtered to obtain synthetic phage using 0.45 μm filter.

## OD measurement

Bacteria overnight cultures were diluted 100-fold in LB media with Cm (34 μg/mL). In 96-well plates, 100 μL of the culture and 2 μL of the phages (MOI $5 \times 10^{-4}$) were incubated at 37 °C at 600 rpm using BioTek LogPhase 600 Microbiology Reader (Agilent Technologies, Inc., Santa Clara, CA, USA). The optical density (OD600) was measured at 10-min intervals for 8 h. Three wells contained only LB media for the background subtraction. Three technical replicate experiments were performed for each assay. The blank values were subtracted from the measured data. The average of three replicates was then calculated and displayed as a line graph, with individual raw data points plotted.

## Plasmid construction method

DNA was amplified using the primers listed in Supplementary Table 2 and KOD One® according to the manufacturer's protocol. Each 50 μL reaction was performed with the following thermal cycling conditions: initial denaturation at 98 °C for 10 seconds, followed by 35 cycles of denaturation at 98 °C for 10 seconds, annealing at 55 °C for 5 seconds, and extension at 68 °C for 5 seconds per kilobase of the target fragment. The amplified fragments were assembled using NEBuilder HiFi DNA Assembly Master Mix. The pKLC83a anti-Tmn plasmid used for identification of anti-Tmn was generated by inserting *ORF35* (*anti-Tmn*) of the ΦSMS22 phage into pKLC83a as a template. The pKLC83a anti-Tmn$^{ΦSMW22, ΦSMA2, ΦKSS9, ΦSMS24, and}$ $^{ΦDruSM5}$ used to evaluate the activity of anti-Tmn homolog were created using pKLC83a as a template and the *anti-Tmn* homologs of each phage was inserted. The Tmn variants, PLG(TmnA-JBABADF-19-0057), PLG(TmnA-JBBDABA-19-0002), PLG(TmnA-JBABADI-19-0057), and PLG(TmnA-JBBEABG-19-0024), used to assess the phage inhibitory effect, were constructed by inserting the Tmn-annotated regions identified using Defense Finder into the PLG001 plasmid. These annotated regions were derived from clinical isolates of 2 *Escherichia coli*, 1 *Enterobacter cloacae*, and 1 *Klebsiella pneumoniae*, respectively. The plasmids constructed in this study are shown in Supplementary Table 4.

## Exploration of Tmn-immune evasion protein

The genomic homology of ΦSMS22 and ΦSMA8 was analyzed using Clustal W. Then, DH10B with and without Tmn expression were cultured overnight and inoculated with these two phages using spot test. Experiments were independently performed in triplicate. The number of plaques formed was counted the following day. The efficiency of plating (EOP) was calculated by dividing the number of plaques observed on Tmn-expressing bacterial lawns by those on bacterial lawns carrying an empty vector. The logarithm of the EOP values was calculated and represented as bar graphs, with error bars indicating standard deviations. Next, ΦSMS22$_{ΔORF35}$, with ORF35 of ΦSMS22 knocked out, was synthesized. ΦSMS22$_{ΔORF35}$ was

named ΦSMS22$_{Δanti-Tmn\_1-3}$, and ΦSMS22 WT and ΦSMS22$_{Δanti-Tmn\_1-3}$ were inoculated with Tmn expressing and Tmn non-expressing bacteria by spot assay and the results were summarized in a bar graph. Moreover, Tmn and anti-Tmn were co-expressed in DH10B {(Tmn,anti-Tmn) = (−,−), (−,+), (+,−), (++)} and cultured overnight. Cells were then inoculated with ΦSMS22 series phage and ΦSMS22$_{Δanti-Tmn}$_1-3 by spot assay. The same operation was performed for T1-T7. The monomeric and dimeric structures of anti-Tmn (*ORF35* from ΦSMS22) were predicted by Alpha-Fold2, respectively.

## Phylogenetic tree construction

For Tmn, all 397 sequences listed on the Defense Finder Wiki were downloaded[22]. Redundant amino acid sequences with identical accession numbers were removed, and sequences with ≥95% identity were clustered, resulting in 81 unique sequences. Homologous protein sequences were aligned using mafft v7.511[36] with default settings. A maximum-likelihood phylogenetic tree was constructed using iq-tree v1.6.12 with the following parameters: -bb 1000 -m TESTNEW -nt AUTO -wbtl -pre[37]. ModelFinder, implemented in iq-tree, identified the best-fitting model (JTT + F + R6).

For anti-Tmn, homologs of the SM-S22 anti-Tmn sequence were identified via BLASTp (E-value cutoff: 1e-5; minimum identity: 30%; coverage threshold: 80%) against the non-redundant protein sequences (nr) database, yielding 75 homologs. The sequences were clustered using MMseqs2 easy-cluster with 95% identity and 95% coverage thresholds. The clustered sequences were aligned using mafft v7.511[36] with default settings, and an HMM profile was built using hmmbuild.

Protein sequences from bacteriophages (2,091,036 sequences) and archaeal viruses (18,046 sequences) were downloaded from GenBank. Homologs were identified using hmmsearch with a GA cut threshold of 20, yielding 85 homologs. Of these, 48 sequences meeting the criteria of profile coverage ≥40% and identity ≤95% were selected for phylogenetic analysis. A phylogenetic tree was constructed as above using the best-fitting model (JTT + G4). The resulting tree was visualized using iTOL[38].

## Variation of inhibitory activity of anti-Tmn homologs against Tmn immunity

The genomic homology of the ΦSMS22 series (ΦSMS22, ΦSMA8, ΦSMW22, ΦSMA2, ΦKSS9, ΦSMS24, ΦDruSM5) was analyzed using Clustal W to generate a phylogenetic tree. Tmn and the anti-Tmn from ΦSMS22 identified in this study, or anti-Tmn homologs from ΦSMW22, ΦSMA2, ΦKSS9, ΦSMS24, and ΦDruSM5, were co-expressed in *E. coli* DH10B. These bacteria were cultured overnight and were infected with the ΦSMS22 series and ΦSMS22$_{Δanti-Tmn\_1-3}$ using spot test and the results were summarized in a bar graph. Furthermore, four Tmn variants cloned from clinical isolates were co-expressed with their anti-Tmn homologs in *E. coli* DH10B. These bacteria were then infected with the ΦSMS22 series, ΦSMS22$_{Δanti-Tmn\_1-3}$, and ΦT1–T7 using spot tests. The results were quantified and presented as bar graphs.

## Synthesis of phages evading Tmn defense system using anti-Tmn

ΦKSS9$_{anti-Tmn}$ was synthesized by replacing the *anti-Tmn* homolog from ΦKSS9 with an *anti-Tmn* derived from ΦSMS22. Bacterial strains, both Tmn-expressing and non-expressing, were cultured overnight and subsequently infected with ΦKSS9 WT and ΦKSS9$_{anti-Tmn\_1-3}$ using spot assays. The outcomes were represented in a bar graph. Also, ΦKSS9 and ΦKSS9$_{anti-Tmn\_1-3}$ were inoculated with Tmn expressing and non- expressing strains in liquid culture and the OD600 was measured. The results are summarized in a line graph.

## Acquisition of phages escaping Tmn immunity

The Tmn expressing strains were grown overnight. ΦSMA8 and ΦSMS22$_{Δanti-Tmn\_1-3}$ were added at an MOI 1 to 100 μL 100-fold diluted of the overnight culture, respectively. The mixture was co-cultured for 24 h and filtered to obtain a phage solution. These phages were tested against

Tmn-expressing and non-expressing bacterial strains using spot assays. The next day, single plaques were picked from the plates and mixed with the Tmn-expressing strain and co-cultured until the medium became clear. ΦSMA8_em and ΦSMS22$_{Δanti-Tmn\_1-3}$_em was obtained by filtering the culture through a 0.45 μm filter. ΦSMA8_em and ΦSMS22$_{Δanti-Tmn\_1-3}$_em was inoculated with Tmn expressing and non-expressing strains using spot test. The mutation sites were identified by reading the sequence of ΦSMA8_em and ΦSMS22$_{Δanti-Tmn\_1-3}$_em and aligning them with WT. Phages in which three mutant sites of ΦSMS22$_{Δanti-Tmn\_1}$_em were returned to WT one by one (ΦSMS22$_{Δanti-Tmn\_1}$_em$_{CDS18C\_1-3}$, ΦSMS22$_{Δanti-Tmn\_1}$_em$_{CDS25C\_1-3}$, ΦSMS22$_{Δanti-Tmn\_1}$_em$_{Nmad5C\_1-3}$). These were inoculated with Tmn expressing and non-expressing strains using spot test and the results were summarized in a bar graph. To further investigate, we synthesized ΦSMS22$_{Δanti-Tmn\_1\_ΔNmad5\_1-3}$ by deleting the *nmad5* from ΦSMS22$_{Δanti-Tmn\_1}$. These phages were subjected to spot tests against Tmn-expressing and non-expressing bacterial strains, and the results were summarized as a bar graph.

### Cloning of *nmad5*

To confirm whether *nmad5* is the Tmn trigger, cloning of the *nmad5* gene was attempted. Using pSC101a as the template, *nmad5$_{WT}$* and *nmad5$_{K296E}$* were cloned from SMS22$_{Δanti-Tmn\_1}$ and ΦSMS22$_{Δanti-Tmn\_1-3}$_em, respectively. As a positive control, the *rfp* gene was also cloned in parallel. All experiments were performed in triplicates (n = 3). Following transformation, colony counts were measured using an Interscience Scan 4000 colony counter set at a sensitivity of 40. The results were summarized in a bar graph.

### Synthesis of phages evading Tmn defense system utilizing *nmad5* mutations

ΦSMS22$_{Δanti-Tmn}$_Nmad5$_{296E\_1-3}$ with Lys296Glu mutation in *nmad5* of ΦSMS22$_{Δanti-Tmn\_1}$ was synthesized. The Tmn expressing and non-expressing bacteria were cultured and were inoculated with ΦSMS22$_{Δanti-Tmn\_1\_}$ and ΦSMS22$_{Δanti-Tmn\_1}$_Nmad5$_{296E\_1-3}$ by spot assay and the results were summarized in a bar graph. Moreover, ΦKSS9 and ΦKSS9$_{anti-Tmn\_1-3}$ were inoculated with Tmn expressing and non-expressing strains in liquid culture and the OD600 was measured. The results are summarized in a line graph.

### Statistics and reproducibility

The spot tests conducted to assess phage infectivity against bacterial strains were performed with biological replicates (n = 3). The results were analyzed using GraphPad Prism 10 (v. 10.3.1), and the mean and standard deviation were calculated and presented as bar graphs with error bars. Raw data points were plotted on the bar graphs. For OD measurements to evaluate phage infectivity in liquid culture, technical replicates (n = 3) were used. The results were analyzed using GraphPad Prism 10 (v. 10.3.1), and the mean values were calculated. Line graphs were generated with raw data points plotted accordingly. Additionally, to account for potential PCR errors, three independent synthetic phage clones were consistently selected for experiments.

### Reporting summary

Further information on research design is available in the Nature Portfolio Reporting Summary linked to this article.

### Data availability

The genome sequence of bacteriophage SM_S22 was deposited in NCBI under accession number PP627283. Whereas the genome data of phages Dru_SM5, KS_S9, SM_A2, SM_A8, SM_S24, and SM_W22 were deposited under accession numbers: PP920550 - PP920555. The PLG026 is available from Addgene under ID 157904, and the sequence of Tmn (JBABADF-19-0057, JBBDABA-19-0002, JBABADI-19-0057, JBBEABG-19-0024) is available under the accession number (PQ687040-PQ697043) in the NCBI GenBank database. Source data for the figures can be found in the supplementary data file.

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

## Acknowledgements

This work was supported by the Japan Agency for Medical Research and Development (Grant No. JP21gm1610002, JP21fk0108496, JP21wm0325022, JP22fk0108532, JP23wm0325065, JP24fk0108698 to K. Kiga; JP22fk0108562 and JP23fk0108599 to K. Chihara; JP23wm0325065 to AH. Azam), and JSPS KAKENHI (Grant No. 21H02110 and 21K19666 to K. Kiga; 23K19475 to K. Chihara; 22K20575 to S. Ojima) and Sasakawa Scientific Research Grant to W. Yamashita. The funders had no role in the study design, data collection and analysis, decision to publish, or preparation of the manuscript.

## Author contributions

W. Yamashita conducted the experiments under the supervision of K. Kiga, interpreted the results, performed bioinformatical analysis and drafted the initial manuscript. K. Chihara and A.H. Azam secured funding, conducted the experiments, performed bioinformatical analysis, interpreted the results, and revised the manuscript. K. Kondo performed bioinformatical analysis and analyzed the data. S. Ojima and A. Tamura provided experimental resources. M. Imanaka conducted the experiments and revised the manuscript. S. Tsuneda, K. Watashi, Y. Takahashi, and N.L. Franklin reviewed the experiments and manuscript. K. Kiga designed the research, secured funding and wrote the manuscript. All authors reviewed and approved the final version of the manuscript.

## Competing interests
The authors declare no competing interests.
