## [Transparent Peer Review file · Communications Biology]

Phage engineering to overcome bacterial Tmn immunity in *Dhillonvirus*

Corresponding Author: Dr Kotaro Kiga

Version 0:

Reviewer comments:

Reviewer #1

(Remarks to the Author)

Bacteria have evolved a myriad of defense systems to overcome phage infection. Facing bacterial immunity, phages have evolved diverse ways to antagonize defense by encoding anti-defense proteins or by evolving to become undetected. While phages are increasingly used in therapy, there is a strong need to better understand the genetic determinants of phage infectivity, and to devise engineering approaches to improve their infective potential. Here, Yamashita et al. focus on the Tmn defense system to investigate such approaches. They use a collection of closely-related phages they previously assembled to identify an anti-Tmn protein encoded in resistant phages and absent in sensitive phages. They show that a Tmn-susceptible phage can overcome defense once equipped with this anti-Tmn protein, showing that phage sensitivity to defense systems can be engineered. Then, they show that phages lacking this anti-Tmn protein can escape defense by evolving a point mutation in a gene called Nmad5 whose suspected function is to modify the phage DNA. Introducing this point mutation in closely-related phages is sufficient for defense escape, so that the authors conclude that phage sensitivity can also be engineered through this second approach. Overall, I enjoyed very much reading this short manuscript of outstanding clarity. The findings are well reported, with raw data shown in the main or supplementary figures, and the phage engineering experiments are compelling. I will support publication once the following points are addressed:

Major comments:

1) My first concern is about the phylogenetic analysis proposed by the authors. I understand that the protein sequence from *E. coli* Tmn was used as a query a blastp search. However, the parameters used by the authors (>80% coverage and >30% identity) do not allow a sensitive search: this is supported by the fact that most hits come from *E. coli*, and that the number of hits is well below the previously reported prevalence of Tmn in prokaryotes (~1.7% based on Gao et al. 2020 and Tesson et al. 2022). I believe that DefenseFinder would return the most accurate list of Tmn homologs in prokaryotes, which seems readily available at <https://defensefinder.mdmlab.fr/wiki/refseq>. I suggest repeating the phylogenetic analysis with this data instead. In addition, I recommend removing identical or near-identical sequences before computing an MSA and tree since redundant sequences can artificially alter the “weight” of each unique sequence (currently most sequences shown on Fig. S1 are almost identical). I have the same comment regarding the search for anti-Tmn genes: the search parameters (>60% coverage and >70% identity) are not sensitive, which is highlighted by the fact that 61/67 hits come from phages infecting *E. coli*. I suggest repeating this analysis with a more sensitive search which can include: i) clustering the reported hits by homology with mmseqs (say, 90% identity), building an HMM profile with the cluster representatives, and using this HMM profile to search these genomes using hmmsearch; or ii) changing the search parameters (low identity and high coverage would actually be more appropriate) and using an iterative search (e.g. psi-blast). Please also mark on the trees the Tmn and anti-Tmn homologs that were experimentally tested.

2) The authors report that other phages from their collection also encode anti-Tmn genes, but that these homologs are much less efficient as the one from Φ SMS22 at blocking the Tmn homolog they used. In the discussion (lines 241-242), they mention that they “confirmed that the inhibitory activity of anti-Tmn was limited to the specific Tmn used”. However, based on the current data, it remains unclear whether the anti-Tmn protein of Φ SMS22 is particularly potent, or if it is just efficient at blocking the Tmn homolog under study. The authors should clone another Tmn homolog from *E. coli*, distantly related to the one they currently use (based on the phylogenetic tree for instance), and test the efficiency of the five anti-Tmn proteins at inhibiting defense. This would shed light on the specificity of anti-Tmn proteins towards Tmn homologs.

3) Phages can also escape Tmn defense by acquiring mutations in the Nmad5 gene. The authors argue that such mutants become undetected, but in my opinion, this remains only one among several hypotheses. In the discussion, they claim that their finding “aligns with observations in other defense systems like Gabija, Septu, Shedu, mzaABCDE, and Druantia type III, where the effectiveness of the defense depends on the phage's DNA modifications”. However, in the cited cases, the DNA modification shields the genome of phage T4 from the effector nuclease activity instead of making the phage undetected, which is the opposite mechanism as proposed here. The fact that Nmad5 is involved in DNA modification also remains a hypothesis. Are there any other DNA-modifying genes in the neighborhood that could support this? A limitation here is that it remains unknown whether the observation mutation in Nmad5 is a gain of function, or a loss of function. I can imagine several hypotheses regarding the phenotype of Nmad5 mutants: i) if Nmad5 mutants are loss-of-function, then these mutants could escape Tmn because it senses DNA modification or the Nmad5 gene product directly. In this case, a Δ Nmad5 mutant should escape defense in a similar fashion; ii) if Nmad5 mutants are gain-of-function, I can imagine two scenarios: Nmad5 could encode an anti-Tmn gene, but only the K296E allele would be efficient at blocking this specific Tmn homolog; or the Nmad5 mutants would gain DNA modification ability, which would block Tmn sensing or abrogate effector activity. In this case, a Δ Nmad5 mutant should not escape defense. Altogether, the authors could obtain significant insights into this by cloning a Δ Nmad5 mutant and testing its sensitivity to Tmn. A complementary experiment would consist in testing whether the WT and K296E mutants are toxic when co-expressed with Tmn. Beyond solving this specific question, these experiments would provide further insights on Tmn, whose defense mechanism remains unknown.

Minor comments:

-The long alpha helices in anti-Tmn look like they could be transmembrane, which would be interesting regarding the domain architecture of Tmn. Does Phobius or tmhmm predict any transmembrane helix?

-Line 153: “inoculated”

-Methods: please clearly state which Tmn homolog was used (with an accession ID).

-Line 345: please provide a reference for AlphaFold2

Reviewer #2

(Remarks to the Author)

This publication describes the identification of an anti-Tmn protein in phage SMS22 from the Dhillonvirus genus that can infect E. coli containing Tmn, therefore overcoming this defence mechanism. The authors used phage engineering to introduce this anti-Tmn gene into a phage which was sensitive to Tmn and found that the phage was able to overcome the Tmn defense system. They also identified a gene which prevented Tmn from detecting phage infection. The article is generally well written, and the findings are novel which would attract a wide audience. However, there are a couple of issues that I believe need to be addressed before publication.

Major Comments:

1. On line 72, the author states ‘phage SMS22 could infect E. coli expressing the Tmn defence system (pLG024), phage SMA8 lacked this ability (Figs. 1b and S2b)’ however, figure S2b only shows the spot assay for phage SMS22 wild type and anti-Tmn knock outs. I also believe this statement to be incorrect, as in figure S2a the spot assays for SMA8 shows ‘lysis from without’ where the phage is still showing signs of injection but no productive infection (no single plaques). Please correct the figure labelling in text and amend text to reflect that SMA8 is displaying lysis from without.
2. The shades of blue on the PFU/mL heatmaps throughout are difficult to interpret. It would be clearer to display the actual PFU/mL on a scatter plot (including replicates). Also, using a PFU/mL of 10⁻¹ to 10⁻³ seems illogical as this would mean that to get 10⁻³ PFU/mL (as some of the heatmaps show) then there is only 1 virus particle in 1L media. However, in the methods there is no mention of concentrating such a large volume of media to be able to detect so few phages.
3. Line 90-93- In the text it states that 67 phages yielded anti-Tmn homologs, however there are 74 phages included in the phylogenetic tree of anti-Tmn homologs (Figure 1d). Please check for correctness.
4. No statistical tests were conducted on the results, therefore the authors must refrain from describing results as ‘significant’ e.g. five phages were significantly inhibited by Tmn

Minor Comments:

1. Italicise genera names throughout e.g. Dhillonvirus
2. Line 66-66 describes a result and would be better featured in results section rather than introduction
3. Line 292- please include thermocycling conditions used
4. Please specify what fragments were assembled in line 294
5. Line 71- how does this figure relate to previous work? also figure 2a comes before 1a in text
6. Figures are not referred to in the text in sequential order, for example figure 2a is referenced first, then 1b

Version 1:

Reviewer comments:

Reviewer #1

(Remarks to the Author)

In the revised version of their manuscript, Yamashita et al. have addressed all my comments and I therefore warmly

recommend publication ! I wish to congratulate the authors for this strong and thorough study.

Please see a few minor comments below :

Figure 1b : please mention what the three deletion mutants in the middle panel refer to (either in the legend or in the main text).

Figure 1d : « Acinetobactor ».

Line 109: the wording “exclusively conserved within Dhillonvirus” might be misleading in the sense that it suggests that it is a core gene of Dhillonviruses. I suggest rephrasing to “exclusively found within Dhillonvirus”.

Line 132: please mention what pKLC026 refers to.

Table 1: Nmad5 is missing in the “gene role” column of CDS7 (Φ SMA8_em).

Line 468: please specify which database was used for BLASTP.

Reviewers' comments:

Reviewer #1 (Remarks to the Author):

Bacteria have evolved a myriad of defense systems to overcome phage infection. Facing bacterial immunity, phages have evolved diverse ways to antagonize defense by encoding anti-defense proteins or by evolving to become undetected. While phages are increasingly used in therapy, there is a strong need to better understand the genetic determinants of phage infectivity, and to devise engineering approaches to improve their infective potential. Here, Yamashita et al. focus on the Tmn defense system to investigate such approaches. They use a collection of closely-related phages they previously assembled to identify an anti-Tmn protein encoded in resistant phages and absent in sensitive phages. They show that a Tmn-susceptible phage can overcome defense once equipped with this anti-Tmn protein, showing that phage sensitivity to defense systems can be engineered. Then, they show that phages lacking this anti-Tmn protein can escape defense by evolving a point mutation in a gene called Nmad5 whose suspected function is to modify the phage DNA. Introducing this point mutation in closely-related phages is sufficient for defense escape, so that the authors conclude that phage sensitivity can also be engineered through this second approach. Overall, I enjoyed very much reading this short manuscript of outstanding clarity. The findings are well reported, with raw data shown in the main or supplementary figures, and the phage engineering experiments are compelling. I will support publication once the following points are addressed:

Major comments:

1) My first concern is about the phylogenetic analysis proposed by the authors. I understand that the protein sequence from *E. coli* Tmn was used as a query a blastp search. However, the parameters used by the authors (>80% coverage and >30% identity) do not allow a sensitive search: this is supported by the fact that most hits come from *E. coli*, and that the number of hits is well below the previously reported prevalence of Tmn in prokaryotes (~1.7% based on Gao et al. 2020 and Tesson et al. 2022). I believe that DefenseFinder would return the most accurate list of Tmn homologs in prokaryotes, which seems readily available at <https://defensefinder.mdmlab.fr/wiki/refseq>. I suggest repeating the phylogenetic analysis with this data instead. In addition, I recommend removing identical or near-identical sequences before computing an MSA and tree since redundant sequences can artificially alter the “weight” of each unique sequence (currently most sequences shown on Fig. S1 are

almost identical). I have the same comment regarding the search for anti-Tmn genes: the search parameters (>60% coverage and >70% identity) are not sensitive, which is highlighted by the fact that 61/67 hits come from phages infecting *E. coli*. I suggest repeating this analysis with a more sensitive search which can include: i) clustering the reported hits by homology with mmseqs (say, 90% identity), building an HMM profile with the cluster representatives, and using this HMM profile to search these genomes using hmmsearch; or ii) changing the search parameters (low identity and high coverage would actually be more appropriate) and using an iterative search (e.g. psi-blast). Please also mark on the trees the Tmn and anti-Tmn homologs that were experimentally tested.

We sincerely appreciate your detailed feedback regarding the phylogenetic analysis. Following your valuable suggestions, we employed the DefenseFinder tool to identify Tmn homologs. This tool allowed us to capture a more comprehensive dataset, revealing a broader diversity of bacterial species beyond *Escherichia coli*, consistent with the previously reported Tmn prevalence in prokaryotes. To address sequence redundancy, we removed sequences with $\geq 95\%$ identity prior to multiple sequence alignment and tree construction. This ensures that each sequence is weighted appropriately in the analysis.

For anti-Tmn, we followed your recommendations by building an HMM profile and searching protein sequences of all bacteriophages and archaeal viruses in GenBank. Initially, we used BLASTp with parameters aimed at enhancing sensitivity (profile coverage $\geq 40\%$ and identity $\leq 95\%$). This yielded 75 homologs, which we clustered using MMseqs2 easy-cluster with thresholds of 95% identity and 95% coverage. While this slightly differs from the suggested 90% identity, we opted for 95% to ensure higher confidence in the cluster representatives. Subsequently, we constructed HMM profiles and conducted an hmmsearch to refine the identification of homologs. Consistent with our initial findings, anti-Tmn homologs were predominantly found in *E. coli* phages. We believe this result reflects biological constraints or biases in the available data rather than a limitation of the search parameters. We hope that these revisions address your concerns and demonstrate our effort to increase sensitivity while maintaining rigor.

2) The authors report that other phages from their collection also encode anti-Tmn genes, but that these homologs are much less efficient as the one from Φ SMS22 at blocking the Tmn homolog they used. In the discussion (lines 241-242), they mention that they “confirmed that the inhibitory activity of anti-Tmn was limited to the specific Tmn used”. However, based on the current data, it remains unclear whether the anti-Tmn protein of Φ SMS22 is particularly potent, or if it is just efficient at blocking the Tmn homolog under study. The

authors should clone another Tmn homolog from *E. coli*, distantly related to the one they currently use (based on the phylogenetic tree for instance), and test the efficiency of the five anti-Tmn proteins at inhibiting defense. This would shed light on the specificity of anti-Tmn proteins towards Tmn homologs.

Thank you for your thoughtful comment regarding the specificity between Tmn and anti-Tmn homologs. To address this important point, we conducted additional experiments as follows. Using the phylogenetic tree, we constructed and the availability of the strains encoding them from our collection of clinical isolates in Japan, we selected four phylogenetically distant Tmn homologs. These included two homologs from *E. coli*, one from *Enterobacter cloacae*, and one from *Klebsiella pneumoniae*. This selection allowed us to evaluate the inhibitory activity of six anti-Tmn homologs against the defensive activity of these newly cloned Tmn homologs. The results of these experiments demonstrated that specific anti-Tmn homologs inhibited the defensive activity of certain Tmn homologs, supporting the concept of specificity between Tmn and anti-Tmn homologs, as you had suggested. We have incorporated these results into the revised manuscript, with the experimental data presented in Figs. S8 and S9, and the detailed findings and implications discussed in Lines 107–122.

3) Phages can also escape Tmn defense by acquiring mutations in the Nmad5 gene. The authors argue that such mutants become undetected, but in my opinion, this remains only one among several hypotheses. In the discussion, they claim that their finding “aligns with observations in other defense systems like Gabija, Septu, Shedu, mzaABCDE, and Druantia type III, where the effectiveness of the defense depends on the phage's DNA modifications”. However, in the cited cases, the DNA modification shields the genome of phage T4 from the effector nuclease activity instead of making the phage undetected, which is the opposite mechanism as proposed here. The fact that Nmad5 is involved in DNA modification also remains a hypothesis. Are there any other DNA-modifying genes in the neighborhood that could support this? A limitation here is that it remains unknown whether the observation mutation in Nmad5 is a gain of function, or a loss of function. I can imagine several hypotheses regarding the phenotype of Nmad5 mutants: i) if Nmad5 mutants are loss-of-function, then these mutants could escape Tmn because it senses DNA modification or the Nmad5 gene product directly. In this case, a Δ Nmad5 mutant should escape defense in a similar fashion; ii) if Nmad5 mutants are gain-of-function, I can imagine two scenarios: Nmad5 could encode an anti-Tmn gene, but only the K296E allele would be efficient at blocking this specific Tmn homolog; or the Nmad5 mutants would gain DNA modification ability, which would block Tmn sensing or abrogate effector activity. In this case, a Δ Nmad5

mutant should not escape defense. Altogether, the authors could obtain significant insights into this by cloning a $\Delta Nmad5$ mutant and testing its sensitivity to Tmn. A complementary experiment would consist in testing whether the WT and K296E mutants are toxic when co-expressed with Tmn. Beyond solving this specific question, these experiments would provide further insights on Tmn, whose defense mechanism remains unknown.

Thank you for your detailed and insightful feedback. We appreciate your observation regarding the cited cases (Gabija, Septu, Shedu, mzaABCDE, and Druantia type III). Accordingly, we have removed the sentences referring to these cases from the Discussion section.

Regarding your suggestion that DNA-modifying genes tend to form clusters in the phage genomes, we examined the genomic context of *nmad5* but did not identify any additional DNA-modifying genes in its vicinity.

To address whether *nmad5*_{K296E} represents a gain-of-function or loss-of-function, we synthesized a $\Delta nmad5$ mutant and tested its sensitivity to Tmn. The results showed that even with the $\Delta nmad5$ deletion, $\Phi SMS22_{\Delta anti-Tmn}$ could not evade Tmn inhibition, indicating that the *nmad5*_{K296E} mutation confers a gain-of-function phenotype.

Next, as you suggested, we attempted to clone *nmad5* and *nmad5*_{K296E} to investigate whether their co-expression with Tmn induces toxicity. However, both *nmad5* and *nmad5*_{K296E} exhibited strong toxicity, which prevented successful cloning (Fig. R1).

We have incorporated these results into the revised manuscript, with the experimental data presented in Figs. S12 and S13, and the details described in Lines 195-199.

Fig. R1 Cloning of *rfp*, *nmad5_{WT}*, and *nmad5_{K296E}*

Minor comments:

-The long alpha helices in anti-Tmn look like they could be transmembrane, which would be interesting regarding the domain architecture of Tmn. Does Phobius or tmhmm predict any transmembrane helix?

Thank you for your valuable perspective. To address your comment, we carefully examined anti-Tmn using both Phobius and TMHMM to predict the presence of transmembrane domains (Figs. R2 and R3). While no transmembrane domains were identified, it is worth noting that the entire anti-Tmn protein was predicted to be non-cytoplasmic.

```

ID    EMOSS_001
FT    DOMAIN          1    154    NON CYTOPLASMIC.
//

```

Fig. R2 The results of the transmembrane domain search for anti-Tmn using Phobius.

```

# WEBSEQUENCE Length: 154
# WEBSEQUENCE Number of predicted TMHs: 0
# WEBSEQUENCE Exp number of AAs in TMHs: 0.01098
# WEBSEQUENCE Exp number, first 60 AAs: 0.00327
# WEBSEQUENCE Total prob of N-in: 0.23662
WEBSEQUENCE    TMHMM2.0    outside    1    154

```

Fig. R3 The results of the transmembrane domain search for anti-Tmn using tmhmm.

-Line 153: "inoculated"

We revised the manuscript accordingly.

-Methods: please clearly state which Tmn homolog was used (with an accession ID).

The Tmn sequence used was cloned from PLG026 and four clinical isolates. PLG026 is available from Addgene under ID 157904, and the sequences of the other Tmn homologs are available in the NCBI GenBank database under accession numbers PQ687040-PQ697043, as stated in the Data availability section (Lines 684–686).

-Line 345: please provide a reference for AlphaFold2

We have cited the relevant reference for AlphaFold (Citation 23)

Reviewer #2 (Remarks to the Author):

This publication describes the identification of an anti-Tmn protein in phage SMS22 from the Dhillonvirus genus that can infect *E. coli* containing Tmn, therefore overcoming this defence mechanism. The authors used phage engineering to introduce this anti-Tmn gene into a phage which was sensitive to Tmn and found that the phage was able to overcome the Tmn defense system. They also identified a gene which prevented Tmn from detecting phage infection. The article is generally well written, and the findings are novel which would attract a wide audience. However, there are a couple of issues that I believe need to be addressed before publication.

Major Comments:

1. On line 72, the author states 'phage SMS22 could infect *E. coli* expressing the Tmn defence system (pLG024), phage SMA8 lacked this ability (Figs. 1b and S2b)' however, figure S2b only shows the spot assay for phage SMS22 wild type and anti-Tmn knock outs. I also believe this statement to be incorrect, as in figure S2a the spot assays for SMA8 shows 'lysis from without' where the phage is still showing signs of injection but no productive infection (no single plaques). Please correct the figure labelling in text and amend text to reflect that SMA8 is displaying lysis from without.

Thank you for your helpful and constructive comments. As you pointed out, the reference to fig. S2b was incorrect, and it should have referred to fig. S1a. We have corrected the figure label accordingly (Line 68). Additionally, regarding the spot assay for SMA8 in fig. S2a, we have clarified in the revised manuscript that, as you noted, no single plaques were observed, indicating a lysis from without phenomenon. This clarification has been added to the manuscript (Line 67).

2. The shades of blue on the PFU/mL heatmaps throughout are difficult to interpret. It would be clearer to display the actual PFU/mL on a scatter plot (including replicates). Also, using a PFU/mL of 10^{-1} to 10^{-3} seems illogical as this would mean that to get 10^{-3} PFU/mL (as some of the heatmaps show) then there is only 1 virus particle in 1L media. However, in the methods there is no mention of concentrating such a large volume of media to be able to

detect so few phages.

Thank you for your valuable feedback regarding the PFU/mL heatmaps and units. We greatly appreciate your suggestions, which helped us improve the clarity and accuracy of our data presentation. Following your suggestion, the data are now presented as bar graphs instead of a heatmap, and scatter plots have been included to display the raw values from the triplicate experiments. Additionally, upon reviewing your comments, we identified an error in the PFU unit calculation. This has been corrected, and we have verified the accuracy of the revised calculations. To improve clarity, we have also replaced the unit PFU/mL with the Efficiency of Plating (EOP), which represents the infection efficiency relative to the control strain harboring the empty vector.

3. Line 90-93- In the text it states that 67 phages yielded anti-Tmn homologs, however there are 74 phages included in the phylogenetic tree of anti-Tmn homologs (Figure 1d). Please check for correctness.

Thank you for pointing out the inconsistency between the figure and the corresponding text regarding the homolog search of anti-Tmn. We have re-conducted the homolog search of anti-Tmn and based on the updated results, we have revised both the figure and the associated text to ensure consistency throughout the manuscript (Fig. 1d and Lines 83-90)

4. No statistical tests were conducted on the results, therefore the authors must refrain from describing results as 'significant' e.g. five phages were significantly inhibited by Tmn

Minor Comments:

Thank you for your helpful comment. As per your suggestion, we have removed the term "significantly" from the manuscript where it was previously used inappropriately.

1. Italicise genera names throughout e.g Dhillonvirus

We have corrected the formatting of genus names throughout the manuscript, ensuring that they are now italicized.

2. Line 66-66 describes a result and would be better featured in results section rather than introduction

Thank you for your constructive comment. We have moved the relevant content from the Introduction section to the Results section (Lines 107-110) to improve the logical flow of the manuscript.

3. Line 292- please include thermocycling conditions used

Thank you for your comment. In response to your suggestion, we have provided the thermocycling conditions used in our experiments in Table S3.

4. Please specify what fragments were assembled in line 294

We have compiled the fragments used in the assembly in Supplementary Table 3.

5. Line 71- how does this figure relate to previous work? also figure 2a comes before 1a in text

Thank you for pointing out the incorrect citation of "Fig. 2a" in the manuscript. The correct citation should have been to "Fig. 1a", and we have made this correction accordingly (Line 66).

6. Figures are not referred to in the text in sequential order, for example figure 2a is referenced first, then 1b

We realized that "Fig. 2a" was incorrectly cited in the manuscript, and the correct reference should have been "Fig. 1a". We appreciate your attention to this detail, which has contributed to improving the clarity and accuracy of our manuscript.

We appreciate the reviewer's careful reading and constructive feedback. In response to the comments, we have made the following revisions:

Reviewers' comments:

Reviewer #1 (Remarks to the Author):

Figure 1b : please mention what the three deletion mutants in the middle panel refer to (either in the legend or in the main text).

>Thank you for your suggestion. The phage numbers in Figure 1c were incorrect and have been corrected, and we have added descriptions of the three variants in lines 540–543.

Figure 1d : « Acinetobactor ».

>Thank you for pointing this out. Spelling errors have been corrected throughout the manuscript.

Line 109: the wording "exclusively conserved within Dhillonvirus" might be misleading in the sense that it suggests that it is a core gene of Dhillonviruses. I suggest rephrasing to

"exclusively found within Dhillonvirus".

> We appreciate your suggestion and have revised "exclusively conserved within Dhillonvirus" to "exclusively found within Dhillonvirus" to prevent any potential misinterpretation.

Line 132: please mention what pKLC026 refers to.

> Thank you for highlighting this issue. The reference to "pKLC26" was incorrect; it has been replaced with "PLG026," and we have clarified the origin of PLG026 in line 103.

Table 1: Nmad5 is missing in the "gene role" column of CDS7 (Φ SMA8_em).

> The omission of "Nmad5" has been corrected by adding it to Table 1.

Line 468: please specify which database was used for BLASTP.

> Thank you for your comment. Additional details regarding the database used for BLASTP analysis have been included in lines 337–338.